# Trends in Outcome of Hematopoietic Stem Cell Transplantation: 5000 Transplantations and 30 Years of Single-Center Experience

**DOI:** 10.3390/cancers15194758

**Published:** 2023-09-28

**Authors:** Ludmila Stepanovna Zubarovskaya, Ivan Sergeevich Moiseev, Maria Dmidrievna Vladovskaya, Natalia Borisovna Mikhailova, Elena Vladislavovna Morozova, Tatyana Alexandrovna Bykova, Yulia Yurievna Vlasova, Olesya Vladimirovna Paina, Ilya Viktorovich Kazantsev, Olga Alexandrovna Slesarchuk, Anna Gennadyevna Smirnova, Anna Alekseevna Osipova, Liliya Vladimirovna Stelmakh, Alexey Yurievich Polushin, Oleg Valerievich Goloshchapov, Maxim Pavlovich Bogomolny, Maria Arkadievna Estrina, Marina Olegovna Popova, Maxim Anatolievich Kucher, Alisa Georgievna Volkova, Alexander Leonidovich Alyansky, Dmitrii Eduardovich Pevtcov, Natalia Evgenievna Ivanova, Elena Vitalievna Babenko, Nikolai Nikolaevich Mamaev, Tatiana Leonidovna Gindina, Alina Alexandrovna Vitrishchak, Alexei Borisovich Chukhlovin, Elena Vladimirovna Semenova, Sergei Nicolaevich Bondarenko, Alexander Dmitrievich Kulagin, Boris Vladimirovich Afanasyev

**Affiliations:** RM Gorbacheva Research Institute, Pavlov University, 197022 Saint-Petersburg, Russiabmt.lymphoma@gmail.com (N.B.M.); ilya_kazantsev@inbox.ru (I.V.K.); dr.annasmirnova@gmail.com (A.G.S.); dr.osipova_aa@mail.ru (A.A.O.); marina.popova.spb@gmail.com (M.O.P.); dr.sergeybondarenko@gmail.com (S.N.B.);

**Keywords:** allogeneic hematopoietic stem cell transplantation, autologous hematopoietic stem cell transplantation, survival over time, trends

## Abstract

**Simple Summary:**

Despite several registry studies on the longitudinal outcomes of hematopoietic cell transplantation (HCT), there is limited information on the major trends in HCT in developing countries. This single-center study evaluates the development of a large transplantation center over 30 years. The analysis includes 5185 transplantations and focuses on major trends in indications over time and changes in outcomes according to the underlying disease. The most significant improvements of survival after autoHCT were observed in Hodgkin’s disease (HR 0.1, 95% CI 0.1–0.3), multiple myeloma (HR 0.4, 95% CI 0.2–0.7) and solid tumors (HR 0.2, 95% CI 0.2–0.4). The most significant improvements in survival after alloHCT were observed for acute myeloid leukemia (HR 0.3, 95% CI 0.1–0.5), acute lymphoblastic leukemia (HR 0.2, 95% CI 0.1–0.5), Hodgkin’s disease (HR 0.1, 95% CI 0.0–0.4), non-Hodgkin’s lymphomas (HR 0.2, 95% CI 0.0–0.6), inborn diseases (HR 0.2, 95% CI 0.2–0.4) and acquired aplastic anemia (HR 0.3, 95% CI 0.2–0.8).

**Abstract:**

In this single-center analysis, we evaluated the trends in 5185 hematopoietic cell transplantations performed between 1990 and 2022. The study group comprised 3237 allogeneic (alloHCT) and 1948 autologous (autoHCT) hematopoietic cell transplantations. In the multivariate analysis, there was an improvement in event-free-survival (EFS) after autoHCT (HR 0.6, 95% CI 0.4–0.7, *p* < 0.0001) due to reduced cumulative incidence of relapse in the last five years (56% in 2010–2014 vs. 38% in 2015–2022). An improvement in EFS after alloHCT over time was observed (HR 0.33, 95% CI 0.23–0.48, *p* < 0.0001), which was due to reduced non-relapse mortality. No difference in cumulative relapse incidence was observed over the last decade for allografted patients. Survival after autoHCT improved in Hodgkin’s disease (HR 0.1, 95% CI 0.1–0.3), multiple myeloma (HR 0.4, 95% CI 0.2–0.7) and solid tumors (HR 0.2, 95% CI 0.2–0.4), while after alloHCT, improvement was observed in acute myeloid leukemia (HR 0.3, 95% CI 0.1–0.5), acute lymphoblastic leukemia (HR 0.2, 95% CI 0.1–0.5), Hodgkin’s disease (HR 0.1, 95% CI 0.0–0.4), non-Hodgkin’s lymphomas and chronic lymphocytic leukemia (HR 0.2, 95% CI 0.0–0.6), inborn diseases (HR 0.2, 95% CI 0.2–0.4) and acquired aplastic anemia with matched related donors and matched unrelated donors (HR 0.3, 95% CI 0.2–0.8).

## 1. Introduction

Several registry studies were published depicting the trends over time in the outcomes of hematopoietic stem cell transplantation (HCT). Most of them focus on allogeneic HCT and demonstrate a reduction in non-relapse mortality (NRM) as the major driver of improved survival [1,2]. For autologous HCT, several studies describe the evolution of outcomes in patients with lymphomas [3,4]. Rarely in these large registry studies is there a subanalysis for certain diseases; thus, the trends in specific diseases require further investigation. Moreover, the registry studies involve centers with modest transplantation activity, where there is significant variability of HCT outcomes [5]. Thus, the analysis of HCT results in highly active transplantation centers remains an important benchmark for comparison. Despite several such benchmarks from developed countries [6,7], there are limited longitudinal data from centers in developing countries, while it is well known that economic healthcare background plays a role in HCT outcomes [8].

In Russia, the state program to support and develop HCT emerged in 1986 after the accident at the Chernobyl nuclear power plant. The consequence of this was the creation of two new specialized departments of HCT, one of which was opened in 1987 in St. Petersburg on the basis of the N.N. Petrov Research Institute of Oncology. The team of this department, led by Boris Afanasyev B.V. [9], subsequently became the basis for the creation of the first university clinic of HCT in Russia at the Pavlov University. Subsequently, in 2007, it developed into the specialized RM Gorbacheva Research Institute, focusing on problems of HCT. The team of the RM Gorbacheva Research Institute was the first to introduce several key transplantation technologies in Russia: in 1991—the first allogeneic HCT for a child, the first autologous peripheral blood stem cell transplantation in 1997, the program for unrelated donor HCT in 2000, the first HCT from a haploidentical donor with ex vivo depletion in 2006 [10,11] and the first transplantation with preimplantation donor selection for a patient with hereditary disease in 2017 [12]. Since the first transplantation in 1990, more than 5000 HCTs have been performed. Since 2013, the RM Gorbacheva Research Institute conducts registered academic trials in the field of HCT [13]. 

Here, we analyze the major trends in more than 5000 autologous (autoHCT) and allogeneic transplantations (alloHCT) performed by the RM Gorbacheva Research Institute team over decades.

## 2. Materials and Methods

The study included data from 5185 patients who underwent HCT from 1990 to 2022. In the study group, autoHCT was performed in 1948 patients and alloHCT in 3237 patients. Allogeneic stem cell transplantations prevailed in the study group since there were many competing autologous centers in Russia, while very few were allogeneic. Thus, allogeneic transplants prevailed in the study population. The detailed characteristics of the group are presented in Table 1. The median follow-up of living patients was 3 years for autoHCT (range 0–32 years) and 4 years for alloHCT (range 0–28 years).

### Clinical Definitions and Statistical Analysis

Time to disease relapse (cumulative incidence of relapse, CIR), non-relapse mortality (NRM), overall survival (OS) and event-free survival (ESV) were defined as the time from transplantation to the event. A five-year time frame was used to analyze OS and EFS in all cases except the longitudinal analysis of the whole autoHCT and alloHCT, where there was no time frame restriction. Disease recurrence was defined as morphological, cytogenetic, radiological or other laboratory evidence of disease with pre-transplant characteristics, or morphological/radiological evidence without additional evaluation for pre-transplant features. Survival was assessed using the Kaplan–Meier method, and the comparison of groups was carried out using a log-rank test. Multivariate survival analysis was performed using the Cox method for OS and EFS. CIR and NRM were considered to be competing events. The comparison of cumulative incidences was carried out using the Gray test and the multivariate analysis of cumulative incidences using the Fine–Gray method. A five year time frame was also used for the NRM and CIR analyses. The results of the multivariate analyses are presented as Forrest plots. The square markers on the graph mark hazard ratios. The horizontal lines mark the confidence intervals of the hazard ratios. The confidence intervals to the right of 1 characterize factors that worsen survival, those to the left improve survival, and those overlapping with 1 have no statistically significant impact on survival. For all comparisons, a confidence level of 0.05 was used.

## 3. Results

### 3.1. General Trends

To simplify the presentation of trends in indications for transplantation and results of transplantation, the patients were divided into those who received transplantation before 2000 (228), from 2000 to 2009 (970), from 2010 to 2014 (1076) and from 2015 to 2022 (2911). There is a progressive increase in transplantation activity: an average of 23 HCTs per year until 2000, 88 HCTs in 2000–2009, 269 in 2010–2014 and 391 in 2015–2022. There was a growing trend in the number of transplantations per year at the RM Gorbacheva Research Institute (Figure 1).

During the existence of the transplantation program, both the ratio of auto and alloHCT, as well as the indications for their implementation, have significantly changed. Prior to 2000, the proportion of autoHCT was 81% and has steadily declined to 34% in the last five years (Figure 2A). The structure of alloHCT also changed according to the donor type. Before 2000, the graft for alloHCT was collected from an MRD in 95% of all cases; in the last five years, the share of related donors was only 18%, unrelated—44%, and haploidentical—38%, which probably reflects an unselected structure of available donors in the Russian Federation (Figure 2B).

The indications for transplantation have also changed. Before 2010, the indications for autoHCT were evenly distributed among acute leukemias, Hodgkin’s lymphoma (HL), non-Hodgkin’s lymphomas (NHL), multiple myeloma (MM) and solid tumors (ST); in recent years, solid tumors in children and MM in adults constituted the majority of autoHCT indications. In the early years, patients with solid tumors were autografted for breast, testicular cancer and germline tumors, while in the later years, it was solely pediatric tumors. The proportion of NHL, HL and, especially, acute leukemia in the structure of indications for autoHCT has been decreasing over the years. On the other hand, there was an expansion of the transplantation program for autoimmune diseases (AID) in recent years, the share of which became 9% in the structure of indications (Figure 2B). 

As for alloHCT, acute lymphoblastic (ALL) and acute myeloid leukemia (AML) were the main indications throughout the operation of the transplant center; however, an increase in the proportion of aplastic anemia (AA) and chronic myeloproliferative neoplasms/myelodysplastic syndrome (MPN/MDS) was observed, which probably reflects the establishment of national reference centers at the RM Gorbacheva Research Institute for these diseases (Figure 2D). In recent years, for the first time, there has been a decrease in the proportion of patients who undergo HCT without a clinical response of the underlying disease. Until 2015, these patients comprised from 18% to 28% of the transplanted population, but after 2015, their proportion decreased to 11% among all indications for transplantation. Of course, the proportion of such patients in the alloHCT group remains higher than in the autoHCT group, even in the last five years (11.4% vs. 2.7%, *p* < 0.0001). As concerning as non-malignant diseases (AA, inborn) are, in spite of the increase in the absolute number of HCTs, this did not effect the proportion of transplants performed. The allografted patients with solid tumors were exclusively represented by neuroblastoma and Ewing sarcoma.

### 3.2. Survival Outcomes

Thirty-year overall survival (OS) in the whole group of autoHCT and alloHCT first transplants in a hematological response was 42% (95% confidence interval [CI] 39–47%), event-free survival (ES)—35% (95% CI 31–40%). Thirty-year OS after autoHCT was 40% (95% CI 33–47%), EFS—34% (95% CI 23–37%). After alloHCT, the overall thirty-year survival was 50% (95% CI 46–54%), event-free—43% (95% CI 39–47%). In both types of HCT, there was a progressive improvement in patient survival. After standard-risk autoHCT, OS was 23% (95% CI 15–32%) before 2000, 30% (95% CI 22–38%) during 2000–2009, 38% (95% CI 28–48%) during 2010–2014 and 69% (95% CI 63–74%) during 2015–2022 (*p* < 0.0001, Figure 3A). After standard-risk alloHCT, OS was 20% (95% CI 9–33%) before 2000, 24% (95% CI 20–28%) during 2000–2009, 34% (95% CI 30–48%) during 2010–2014 and 56% (95% CI 52–59%) during 2015–2022 (*p* < 0.0001, Figure 3B).

#### 3.2.1. Autologous HCT

The improvement in the results for autoHCT was associated with a decrease in a disease recurrence rate: 60% (95% CI 52–67%) after transplantation before 2000, 57% (95% CI 52–62%) in 2000–2009, 56% (95% CI 49–61%) in 2010–2014, and 38% (95% CI 34–43%) in 2015–2022. Only the reduction in CIR during the last five years was significant (*p* < 0.0001) with no pairwise difference in previous years (*p* > 0.05). NRM after autoHCT was low, regardless of the period of the procedure, and averaged 6% (95% CI 5–8%) with no difference between time periods (*p* > 0.05). The results of autoHCT for the underlying disease without clinical response (salvage autoHCT) did not significantly differ according to the time of the procedure (*p* = 0.74). The OS for this group was 28% (95% CI 14–44%), and EFS was 11% (95% CI 1–33%).

#### 3.2.2. Allogeneic HCT

The survival improvement after alloHCT was associated with a decrease in NRM, which was 59% (95% CI 39–74%) after HCT before 2000, 38% (95% CI 33–43%) in 2000–2009, 23% (95% CI 20–27%) in 2010–2014, 16% (95% CI 14–18%) in 2015–2022 (*p* < 0.0001). The CIR after alloHCT has not significantly decreased in recent years and was the lowest until 2000 due to patient selection: 17% (95% CI 6–33%) after transplantation before 2000, 33% (95% CI 27–38%) in 2000–2010, 38% (95% CI 33–42%) in 2010–2014 and 31% (95% CI 28–34%) in 2015–2022 (*p* = 0.0032).

AlloHCT was performed in 536 patients without a hematological response in the underlying disease (salvage HCT). The survival rate in this group before 2000 was 0%, 5% (95% CI 2–12%) in 2000–2009, 7% (95% CI 4–12%) in 2010–2014 and 17% (95% CI 9–27%) in 2015–2022 (*p* < 0.0001). Event-free survival in the general group of patients from the “rescue” group was quite low—4% (95% CI 1–9%). In addition to “rescue” transplants, there is a trend towards an increased number of repeated alloHCTs. Their number was 0% before 2000, 15% in 2000–2009, 21% in 2010–2014 and 17% in 2015–2022. The main indications for repeated alloHCT were graft failure after the first one and recurrence of the underlying disease. In the cases of repeated transplantation in remission of the underlying disease, the 5-year OS was 38% (95% CI 31–40%), in the case of disease progression—10% (95% CI 5–17%) (*p* < 0.0001).

A decrease in NRM was observed with all types of matched-donor alloHCT. In transplantation from MRD, NRM was 61% (95% CI 38–78%) before 2000, 34% (95% CI 24–45%) in 2000–2009, 22% (95% CI 14–30%) in 2010–2014 and 9% (95% CI 6–14%) in 2015–2022. When transplanted from 9–10/10 HLA-matched MUD, NRM was 53% (95% CI 45–60%) in 2000–2009, 33% (95% CI 28–39%) in 2010–2014 and 16% (95% CI 6–14%) in 2015–2022. Given the use of fairly effective transplant technologies for haploidentical transplantations from the beginning of this program, no significant reduction in NRM was observed for this type of allograft: 30% (95% CI 11–36%) in 2000–2009, 22% (95% CI 11–36%) in 2010–2014 and 23% (95% CI 19–27%) in 2015–2022 (*p* = 0.7). 

#### 3.2.3. Multivariate Analysis of 5-Year Outcomes of HCT

In the alloHCT group, comprising 3237 patients, the following factors adversely affected 5-year OS: the diagnosis of a malignant disease (hazard ratio [HR] 1.5, 95% CI 1.2–1.9, *p* = 0.0003), age ≥18 years (HR 1.8, 95% CI 1.6–2.1, *p* < 0.0001), second transplantation (HR 1.8, 95% CI 1.6–2.1, *p* < 0.0001), non-responsive disease before HCT (HR 2.3, 95% CI 2.1–2.6, *p* < 0.0001), and haploidentical donor (HR 1.8, 95% CI 1.6–2.1, *p* < 0.0001). At the same time, OS after transplantation from MRD did not differ significantly from the results of MUD (HR 1.1, 95% CI 0.9–1.3, *p* = 0.11). Over the years, the multivariate analysis also noted a progressive improvement in OS rates, with the most favorable results in 2015–2022 (HR 0.2, 95% CI 0.2–0.3, *p* < 0.0001) (Figure 4A). For the 5-year EFS, the same factors had a significant effect. A clear improvement in EFS over time was observed (HR 0.33, 95% CI 0.23–0.48, *p* < 0.0001). It may be noted that MUD results were also equivalent to the MRD (HR 1.0, 95% CI 0.9–1.1, *p* = 0.91) (Figure 4B).

In the autoHCT group, comprising 1948 patients, the most significant determinant of EFS was the diagnosis of the underlying disease. Compared to HL as reference, the outcome was worse in ALL (HR 1.5, 95% CI 1.0–2.1, *p* = 0.02) and ST (HR 1.5, 95% CI 1.2–1.8, *p* = 0.0007) and better in NHL (HR 0.6, 95% CI 0.5–0.8, *p* = 0.0005) and MM (HR 0.8, 95% CI 0.6–1.0, *p* = 0.03). Comparable results for EFS were observed in AML (HR 1.2, 95% CI 0.9–1.6, *p* = 0.18). The only factor that affected the EFS in autoHCT was the lack of a clinical response at the time of transplantation (HR 1.7, 95% CI 1.2–2.3, *p* = 0.0019). There was also an improvement in EFS in 2015–2022 (HR 0.6, 95% CI 0.4–0.7, *p* < 0.0001, Figure 4C). The results of autoHCT in autoimmune diseases were not included in the survival analysis, because OS in this group was 98%.

### 3.3. Trends in Specific Diseases

#### 3.3.1. Allogeneic HCT

In AML after alloHCT, there is a progressive improvement in survival, with the best rates in the last five years (HR 0.3, 95% CI 0.1–0.5, *p* < 0.0001). In ALL, a similar trend in improved survival was observed (HR 0.2, 95% CI 0.1–0.5, *p* = 0.0004). In aplastic anemia (AA), the OS after alloHCT from matched related and unrelated donors improved significantly (HR 0.3, 95% CI 0.2–0.8, *p* = 0.0152), mainly due to progress in unrelated donor HCT. However, due to the moderate outcomes of haploidentical HCT in heavily pretreated AA patients in the last 5 years, the OS in the entire AA cohort did not change significantly (HR 0.3, 95% CI 0.1–1.2, *p* = 0.08). In chronic myeloid leukemia (CML), there was no improvement in transplant outcomes over time (HR 0.5, 95% CI 0.2–1.3, *p* = 0.13). Some improvement in the results of alloHCT was observed in myelodysplastic syndrome (MDS) and chronic myeloproliferative neoplasms (MPN) but only in the last five years (HR 0.6, 95% CI 0.3–0.9, *p* = 0.03). A dramatic improvement in OS was observed with alloHCT in HL (HR 0.1, 95% CI 0.0–0.4, *p* = 0.0007), NHL and chronic lymphocytic leukemia (CLL) (HR 0.2, 95% CI 0.0–0.6, *p* = 0.0020). The same pattern was observed in inborn diseases (ID) (HR 0.2, 95% CI 0.2–0.4, *p* < 0.0001) (Figure 5). The best results of alloHCT in the last five years were observed in HL (OS 84%), NHL and CLL (OS 86%), AA (OS 85%) and ID (OS 87%).

#### 3.3.2. Autologous HCT

Interestingly, despite a significant decrease in the number of autoHCTs in acute leukemia, there was an improvement in the results of HCT, reflecting the current practice of autoHCT in molecular remissions (HR 0.4, 95% CI 0.2–0.7, *p* = 0.0038). The autoHCT results also improved in HL (HR 0.1, 95% CI 0.1–0.3, *p* < 0.0001), multiple myeloma (HR 0.4, 95% CI 0.2–0.7, *p* = 0.0037) and solid tumors (HR 0.2, 95% CI 0.2–0.4, *p* < 0.0001). In NHL there were no significant improvements in the results for autoHCT (HR 0.5, 95% CI 0.3–1.2, *p* = 0.15) (Figure 6).

## 4. Discussion

The results of the study demonstrate the kinetics of transplantation center growth in a developing country with a significant concentration of HCT technology in selected centers. The results of the analysis show an important trend with better results for HCT in almost all diseases, which increases the economic efficiency of this method. The trend of improving results is not unique to the RM Gorbacheva Research Institute. Thus, an analysis of the registry of the European Society for Blood and Marrow Transplantation (EBMT) shows that with alloHCT, by 2016, NRM decreased from 30% to 12% after MRD allografts [1]. The decrease in NRM is primarily associated with improved infection control; the use of new antibacterial, antifungal and antiviral drugs; and the increased use of peripheral stem cells as a source of transplant [14]. Second, the use of new methods of graft-versus-host disease (GVHD) prevention has reduced the incidence of the acute form of this complication [15,16]. Third, effective methods for the treatment of acute and chronic GVHD have appeared, which reduce mortality in the event of the development of these formidable complications [17,18]. Finally, screening programs for late complications can reduce the risk of NRM in the late period after HCT [19]. In the case of autoHCT, the opposite situation is observed: NRM remains at a low level, but the relapse incidence is reduced by achieving a better response of an underlying malignant disease before autograft [3]. The comparability of results and trends with the registry data was recently analyzed in our large study, where it was shown that the results are similar in most cases when using modern pre- and post-transplantation therapy [20].

The differences in results arise in the early years of the RM Gorbacheva Institute alloHCT activity. NRM, in the first six months after HCT, was comparable to the published data. However, we documented a relatively high incidence of late NRM beyond 6 months of transplantation. On the contrary, many centers in developed countries demonstrate late NRM not exceeding 5% [1,2]; in our cohorts before 2005, these events contributed to half of NRM after alloHCT. Several explanations could be provided to explain these differences. The most obvious one is that after 6 months, patients were discharged to the general health care system, which, at that time period, failed to treat breakthrough infections, which are the most common cause of NRM after alloHCT [21]. This is additionally complicated by vast distances and long travel time from certain residences. It sometimes takes several days to travel to a transplantation center. The important conclusion is that the establishment of alloHCT program in a developing country requires quick access to a transplant center for the treatment of late complications, either in the form of hospitalization or telemedical consult for all patients. Furthermore, since we analyzed all indications at once, some of them with worse prognoses, such as AML, MDS and MPN, might have negatively skewed the results.

The main trends in indications for alloHCT also largely correspond to the global ones. The number of transplantations for CML has significantly decreased due to the availability of at least five lines of conservative therapy. The proportion of patients with CLL is also decreasing, and there is a decline in transplantation activity in HL due to the emergence of effective drugs. At the same time, the proportion of patients with AML, MDS and MPN is increasing, accounting for up to 57% of all alloHCT recipients [22]. In contrast to the general transplant landscape, the RM Gorbacheva Research Institute has an active autoHCT program in pediatric solid tumors. If, in the early years, most of the patients with solid tumors were adults, with breast cancer and several others, then in recent years, only brain tumors and soft tissue tumors in children remained as indications [23]. The revealed tendency to reduce the proportion of acute leukemias as an indication for autoHCT is typical for most centers [24]. Interestingly, extremely favorable progression-free survival rates have been observed in recent years, especially in AML, which probably reflects the use of such transplants in molecular remissions (MRD-). The latest GIMEMA study shows comparable results of alloHCT and autoHCT in patients who achieve MRD-remission. Thus autoHCT may still be considered in a subset of patients with AML without a donor [25].

An important result of this analysis is the equivalence of the long-term results of matched unrelated and matched related HCTs. These results confirm the previously published EBMT and CIBMTR data HCT [26]. Given the share of MRD in the structure of transplants at 18% over the past five years in the absence of significant patient selection, this figure probably reflects the real current demographic situation. At the same time, the results of haploidentical transplantation turned out to be non-equivalent to the matched related and unrelated allografts. Despite a significant number of studies with similar results for compatible and haploidentical HCTs [26,27], nevertheless, the groups in these studies are not comparable in terms of transplantation technology. The comparison of patients with only post-transplant cyclophosphamide prophylaxis in the latest CIBMTR study shows better patient survival after unrelated HCT [28]. Moreover, modern prevention technologies largely eliminate the differences between 9/10 and 10/10 HLA-matched unrelated donors [29,30] which increases the chances of finding a suitable unrelated donor.

It should be highlighted that we identified significant improvements in HCT results in a number of diseases. These are, first of all, HL, CLL, NHL and MM, i.e., diseases where effective targeted therapy has appeared [31,32,33]. In these diseases, HCT provides an effective consolidation of a deep complete response and allows the achievement of very favorable results in terms of progression-free survival. The most striking example of this success is the results of second-line therapy for HL with the inclusion of checkpoint inhibitors and high-dose chemotherapy followed by autoHCT. Despite the short follow-up period after such programs, it is clear that they can approach the effectiveness of the primary treatment [33,34]. With respect to alloHCT, it is currently known that a number of targeted drugs not only allow a patient in good status to undergo HCT but also have an immunomodulatory effect on the graft-versus-tumor effect or reduce the risk of complications [35,36]. On the other hand, no improvement in outcomes for CML is easily explained by different characteristics of CML patients, where the first chronic phase is largely replaced by patients with advanced disease [37]. In the case of aplastic anemia, on the contrary, upfront MRD allograft remains the therapy of choice in young patients with excellent long-term outcomes; as well, MUD HCT immediately after IST failure provides very promising outcomes [38]. These two examples from our study indicate that such a broad analysis cannot be used for decision-making or clinical recommendations, because it lacks a deep evaluation of the disease biology, status, clinical course and patient status at transplant. Moreover, an obvious limitation of the study is a lack of details on the patients’ characteristics, diseases, conditioning regimens, early and late complications and other generally accepted details of HCT. Nonetheless, it gives a glance at the trends that allow for planning the development of existing and new transplantation programs.

## 5. Conclusions

This analysis of more than 5000 transplantations showed that hematopoietic stem cell transplantation is a dynamically developing area of medicine with a dramatic improvement in clinical outcomes. The most significant improvements in outcomes were observed in diseases where effective targeted non-toxic drugs have appeared. The future of this field of medicine can be predicted as a combination of targeted and cellular therapies, causing the gradual decline of classical chemotherapy and high-dose preparation before transplantation.

## Figures and Tables

**Figure 1 cancers-15-04758-f001:**
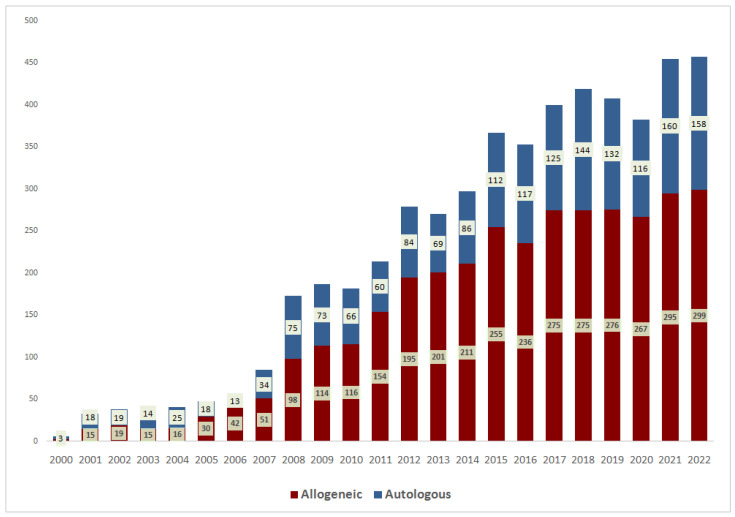
Number of hematopoietic stem cell transplantations at the RM Gorbacheva Research Institute by year of transplantation.

**Figure 2 cancers-15-04758-f002:**
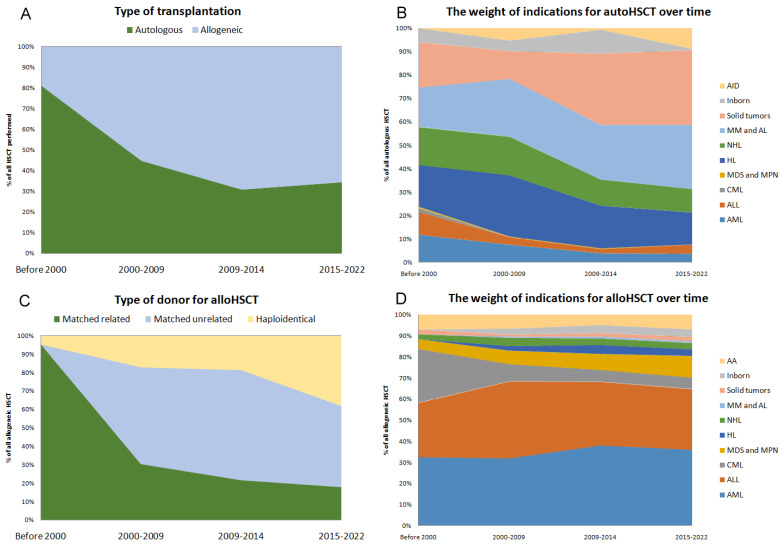
The number of hematopoietic stem cell transplantations at RM Gorbacheva Research Institute by year of transplantation Changes in the transplant program at RM Gorbacheva Research Institute over time. (**A**) The ratio of autoHCT and alloHCT. (**B**) Indications for autoHCT. (**C**) The ratio of donor types for alloHCT. (**D**) Indications for alloHCT.

**Figure 3 cancers-15-04758-f003:**
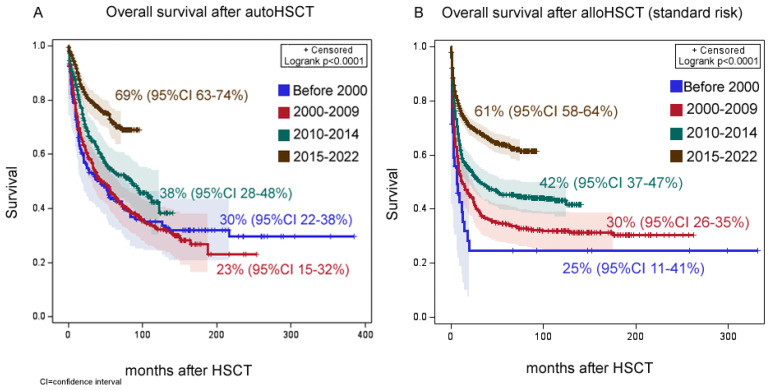
The impact of year of transplant on overall survival. (**A**) Overall survival after autoHCT. (**B**) Overall survival after alloHCT.

**Figure 4 cancers-15-04758-f004:**
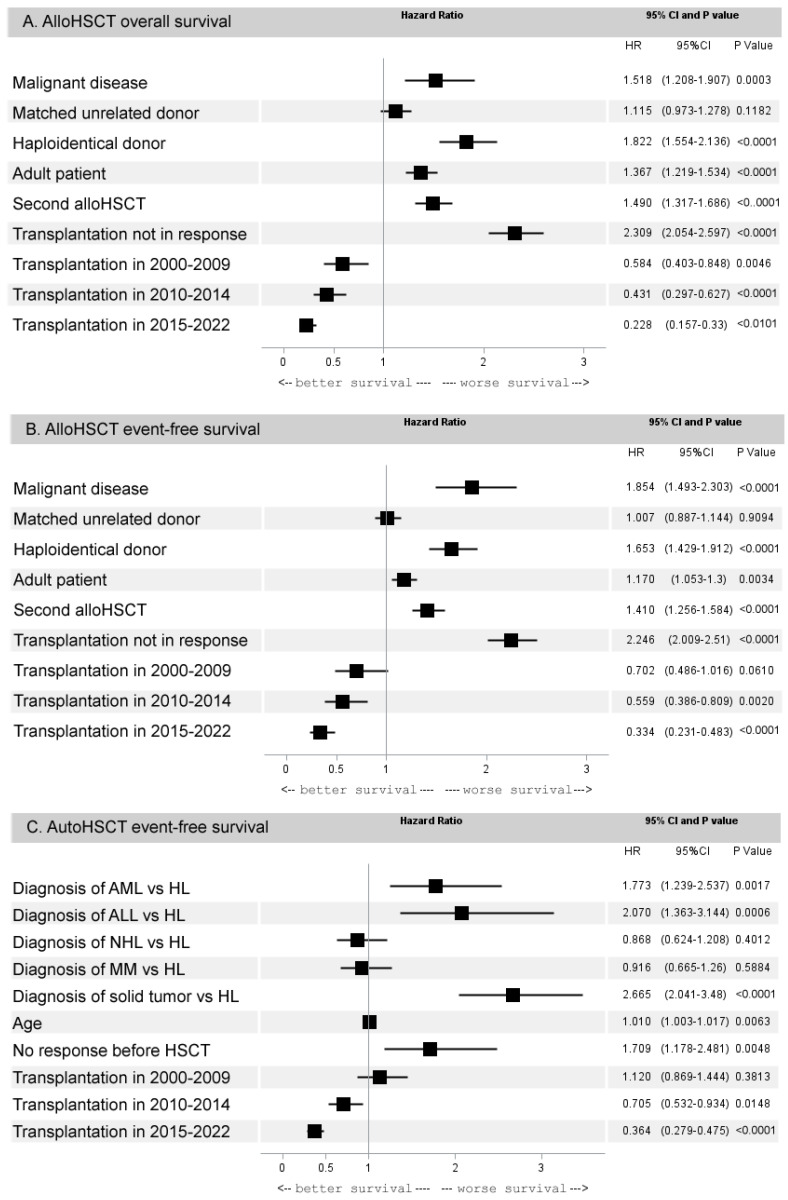
Forest plot with results of multivariate analysis. (**A**) Influence of factors on overall survival after alloHCT. (**B**) Influence of factors on event-free survival after alloHCT. (**C**) Influence of factors on event-free survival after autoHCT.

**Figure 5 cancers-15-04758-f005:**
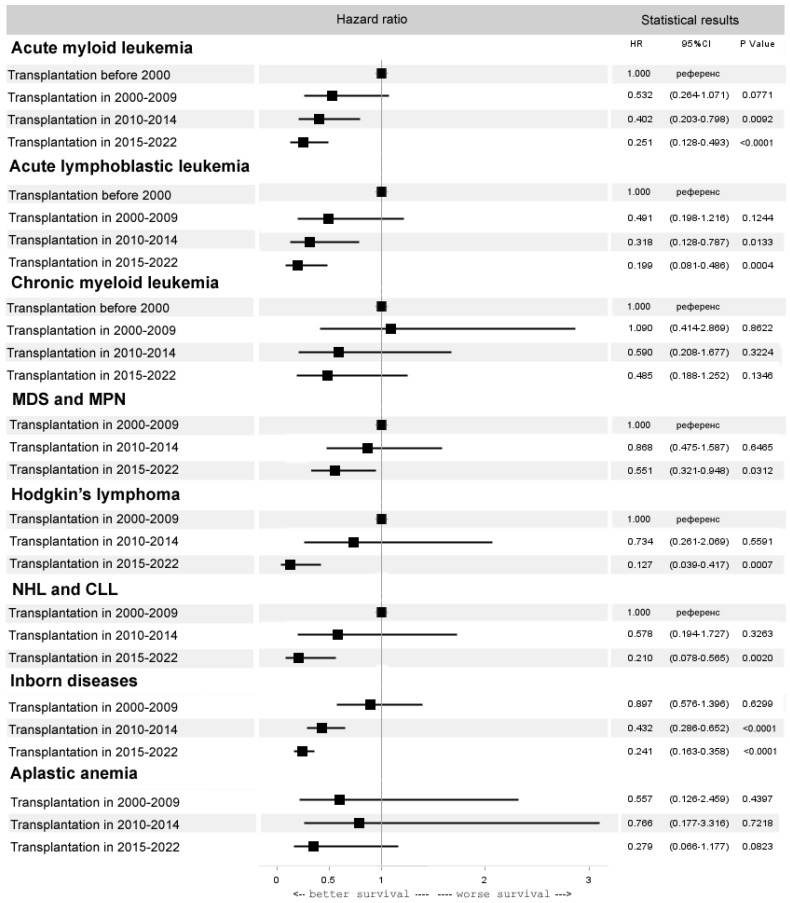
Forest plot of hazard ratios for 5-year overall survival after alloHCT for selected diseases.

**Figure 6 cancers-15-04758-f006:**
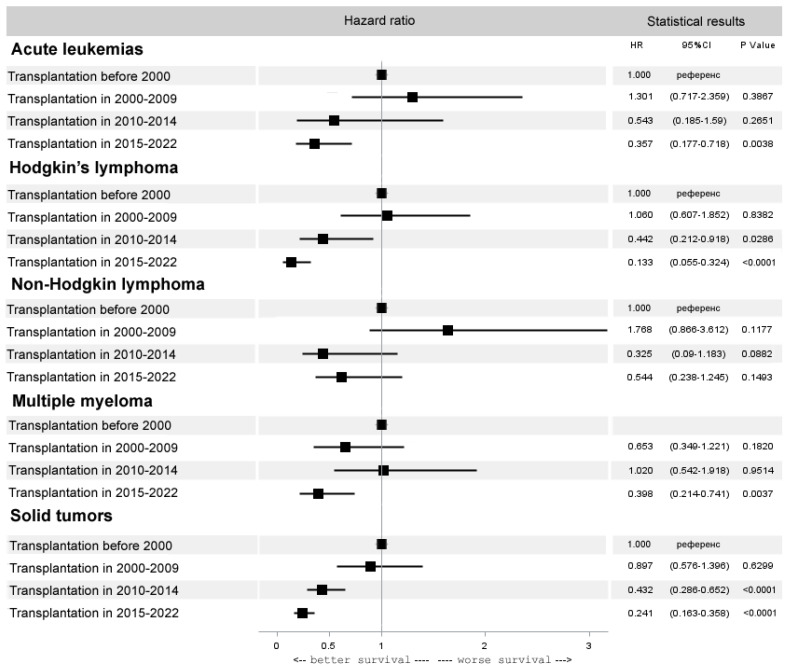
Forest plot of hazard ratios for 5-year overall survival after autoHCT for selected diseases.

**Table 1 cancers-15-04758-t001:** Characteristics of transplantations performed in 1990–2022.

	Autologous, n (%)	Allogeneic, n (%)
Number	1948 (38)	3237 (62)
Age		
Median (range), years	30 (0–71)	23 (0–77)
Children	636 (32.7)	1190 (36.8)
Adults	1312 (67.3)	2047 (63.2)
Diagnosis		
Acute myeloid leukemia	105 (5.4)	1159 (35.8)
Acute lymphoblastic leukemia	77 (4.0)	977 (30.2)
Chronic myeloid leukemia	4 (0.2)	201 (6.2)
Myeloproliferative diseases and myelodysplastic syndrome	3 (0.2) *	302 (9.3)
Hodgkin’s lymphoma	341 (17.5)	107 (3.3)
Non-Hodgkin’s lymphomas Chronic lymphocytic leukemia	238 (12.2) 6 (0.3)	63 (1.9) 40 (1.2)
Multiple myeloma	488 (25.0)	17 (0.5)
Solid tumors	568 (29)	65 (2.0)
Aplastic anemia	1 (0.1) **	210 (6.5)
Congenital diseases	1 (0.1) ***	96 (3.0)
Autoimmune diseases	116 (6.0)	0 (0)
Number of transplantations for single patient		
1st HCT	1738 (89.2)	2655 (82.0)
2nd HCT	209 (10.7)	499 (15.4)
3rd HCT	1 (0.1)	77 (2.4)
4th HCT	0 (0)	6 (0.2)
Donor		
Matched related	NA	707 (21.8)
Matched unrelated	NA	1568 (48.4)
Mismatched related	NA	962 (29.7)
Indications for transplantation		
Standard-risk HCT	1886 (96.8)	2701 (83.4)
Salvage HCT ****	62 (3.2)	536 (16.6)

The percentages are given according to the total number of patients with different types of HCT. *—patients in MDS CR after chemotherapy in the early period, **—syngeneic HCT, ***—congenital histiocytosis, ****—salvage transplantation refers to “in development” indications in recommendation to HCT.

## Data Availability

The data presented in this study are available on request from the corresponding author.

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
