# Peer review of "Trends in Outcome of Hematopoietic Stem Cell Transplantation: 5000 Transplantations and 30 Years of Single-Center Experience"

_cancers, 2023, doi:10.3390/cancers15194758_

Round 1

Reviewer 1 Report

This paper summarizes the work done at a single transplant center in Russia

over a 30 year period and reflects the progress achieved worldwide.

It is an interesting document for all researchers working in the field.

Minor points: line 170-172 not clear to me.

                       line 213 :interesting...may be just noted

                       page 8/18 not in response -not responding

Author Response

Dear Reviewer, 

Thank you your time and expertise! The following changes were applied according to your comments: 

Minor points: line 170-172 not clear to me. - Sentence re-written.

                       line 213 :interesting...may be just noted - Sentence re-phrased.

                       page 8/18 not in response -not responding - Sentence re-phrased.

Reviewer 2 Report

Zubarovskaya L.S., et al. gave a well investigation of retrospective data of 5185 hematopoietic stem cell transplantations in a single center. It showed the trends of hematopoietic stem cell transplantation including the numbers of transplantation, the indications of transplantation, the outcomes of transplantation and the classification of transplantations in highly active transplantation centers. It was nearly the history of clinical treatment of hematopoietic stem cell transplantation in the last three decades. We can see the change of disease composition for specific transplantations, the progression of survival, relapse, NRM outcomes in the last three decades. We can see the change of transplantation is connect with the develop of new treatment regimens, targeted drugs, immune therapy and the improvement of technologic support. We can also predict the trends of transplantation in the future from this study. It was a valuable retrospective study constituted a large population. It was also a well written research paper. I have no further comments.

Author Response

Dear reviewer, 

thank you for your time and expertise reviewing the manuscript! 
Since there was no suggested corrections, no further changes were made to the manuscript.

Reviewer 3 Report

The authors have done marvelous job in collecting the details about transplant trends over three decades in single institution study. I have following comments about the study.

-- In the CIMBTR studies, autologous transplant are numerically higher than allogeneic transplants, what is possible reason for having less autologous transplant.

-- What were the solid tumors that got autologous transplant, were they mostly testicular cancer, or breast cancer as they were done in 1990s? In addition there are 65 patients who got allogeneic transplant for solid tumors, how is that a thing?

-- Why does the graph starts from year 2000 when you have patients before 2000 as well, May be representing pre-2000 patients be a good idea.

-- For survival analysis, lumping all the patients together, including benign and malignant patients might represent a skew, which is addressed by doing analysis with each major disorder like AML, ALL, MDS, MM, AA, AID separately. Please include in the discussion about that as well. 

Author Response

Dear reviewer,

thank you for the time and expertise! According to your comments the following changes were made: 

-- In the CIMBTR studies, autologous transplant are numerically higher than allogeneic transplants, what is possible reason for having less autologous transplant. - information added that there were competing autologous centers, but not allogeneic which is the reason for prevailing alloHCTs in the study population. 

-- What were the solid tumors that got autologous transplant, were they mostly testicular cancer, or breast cancer as they were done in 1990s? In addition there are 65 patients who got allogeneic transplant for solid tumors, how is that a thing?  - Information added that both for auto and allo the indications included only pediatric tumors in recent years. For allo it were neuroblastomas and Ewing. 

-- Why does the graph starts from year 2000 when you have patients before 2000 as well, May be representing pre-2000 patients be a good idea. - for some indications there were no transplants  before 2000 in our center, for those which were transplantated we included these first transplantations as reference on forrest plot (marked as 1).

-- For survival analysis, lumping all the patients together, including benign and malignant patients might represent a skew, which is addressed by doing analysis with each major disorder like AML, ALL, MDS, MM, AA, AID separately. Please include in the discussion about that as well. - information about drawbacks of analyzing all indications at once added to discussion. The trends in each disease are summarized in figure 6. The article is already on top of the word limit to discuss them separately in detail.